# Synergistic Antimicrobial Effect of Cold Atmospheric Plasma and Redox-Active Nanoparticles

**DOI:** 10.3390/biomedicines11102780

**Published:** 2023-10-13

**Authors:** Artem M. Ermakov, Vera A. Afanasyeva, Alexander V. Lazukin, Yuri M. Shlyapnikov, Elizaveta S. Zhdanova, Anastasia A. Kolotova, Artem S. Blagodatski, Olga N. Ermakova, Nikita N. Chukavin, Vladimir K. Ivanov, Anton L. Popov

**Affiliations:** 1Hospital of the Pushchino Scientific Center of the Russian Academy of Sciences, 142290 Pushchino, Russiava_vera_afanaseva@mail.ru (V.A.A.); dla_lisa@mail.ru (E.S.Z.); 2Institute of Theoretical and Experimental Biophysics of the Russian Academy of Sciences, 142290 Pushchino, Russia; yuri.shlyapnikov@gmail.com (Y.M.S.); anas.kolotowa2010@ya.ru (A.A.K.); bswin2000@gmail.com (A.S.B.); knopsik-svetik@yandex.ru (O.N.E.); chukavinnik@gmail.com (N.N.C.); 3ANO Engineering Physics Institute, 142210 Serpukhov, Russia; 4Troitsk Institute of Innovative and Thermonuclear Research (JSC “SSC RF TRINITY”), 108840 Moscow, Russia; lazukin_av@mail.ru; 5Scientific and Educational Center, State University of Education, 105005 Moscow, Russia; 6Kurnakov Institute of General and Inorganic Chemistry, Russian Academy of Sciences, 119991 Moscow, Russia; van@igic.ras.ru

**Keywords:** cold argon plasma, cerium oxide, cerium fluoride, tungsten oxide, nanoparticles, bacteria, antimicrobial effect, combined treatment

## Abstract

Cold argon plasma (CAP) and metal oxide nanoparticles are well known antimicrobial agents. In the current study, on an example of *Escherichia coli*, a series of analyses was performed to assess the antibacterial action of the combination of these agents and to evaluate the possibility of using cerium oxide and cerium fluoride nanoparticles for a combined treatment of bacterial diseases. The joint effect of the combination of cold argon plasma and several metal oxide and fluoride nanoparticles (CeO_2_, CeF_3_, WO_3_) was investigated on a model of *E. coli* colony growth on agar plates. The mutagenic effect of different CAP and nanoparticle combinations on bacterial DNA was investigated, by means of a blue–white colony assay and RAPD-PCR. The effect on cell wall damage, using atomic force microscopy, was also studied. The results obtained demonstrate that the combination of CAP and redox-active metal oxide nanoparticles (RAMON) effectively inhibits bacterial growth, providing a synergistic antimicrobial effect exceeding that of any of the agents alone. The combination of CAP and CeF_3_ was shown to be the most effective mutagen against plasmid DNA, and the combination of CAP and WO_3_ was the most effective against bacterial genomic DNA. The analysis of direct cell wall damage by atomic force microscopy showed the combination of CAP and CeF_3_ to be the most effective antimicrobial agent. The combination of CAP and redox-active metal oxide or metal fluoride nanoparticles has a strong synergistic antimicrobial effect on bacterial growth, resulting in plasmid and genomic DNA damage and cell wall damage. For the first time, a strong antimicrobial and DNA-damaging effect of CeF_3_ nanoparticles has been demonstrated.

## 1. Introduction

Cold argon plasma (CAP) is ionised gas which temperature is comparable to a physiological value (30–40 °C), whereas so-called thermal plasma used in industrial applications has a temperature of 3000–5000 °C, having a devastating effect on biological objects [1]. In 1996, Laroussi et al. first demonstrated that glow discharge plasma generated at atmospheric pressure can be a good alternative for a sterilisation process [2]. Its microbiocidal effect is achieved via damaging and killing bacteria by reactive oxygen species (ROS), reactive nitrogen species (RNS), charged particles, UV photons, generated in the plasma flow [3,4].

In vitro and in vivo studies have shown that CAP has no allergic, toxic or mutagenic effects and is safe for medical applications [5]. The most common applications of cold plasma are regenerative medicine (wound healing) and cancer treatment [6]. The presence of conditionally pathogenic or pathogenic microorganisms in a wound often leads to prolongation of its healing period. CAP is effective in the sterilisation of gram-positive and gram-negative bacteria, biofilms, viruses and fungi [1]. Thus, cold argon plasma is used in medicine for reducing wound healing duration due to the destruction of microorganisms, thus improving postoperative quality of life [7,8].

Metal oxide nanoparticles have been repeatedly tested for their antimicrobial activity. It has been shown that they are able to inactivate most pathogens, in various environments. Among metal oxides, cerium oxide nanoparticles (CeO_2_ NPs) are uniquely able to act as antioxidants, due to the transition between Ce^3+^ and Ce^4+^ states [9]. CeO_2_ NPs also show an ability to increase the efficiency of existing antimicrobials, and thus they could be used as antibiotic adjuvants against drug-resistant pathogens. The synergistic antibacterial effect is the result of the interaction of CeO_2_ NPs with bacterial outer membranes [10]. It has been demonstrated that CeO_2_ NPs can destroy bacterial biofilms [11]. CeO_2_ NP-modified surfaces can prevent the adhesion, proliferation and spread of *Pseudomonas aeruginosa* [12]. Additionally, it has been shown that CeO_2_ nanorods have an antimicrobial action through haloperoxidase activity. Enzyme-like activities of cerium-based nanorods vary with differences in Ce^3+^ concentration, oxygen vacancy concentration and band gap energy [13]. 

Given that the presence and concentration of Ce^3+^ on the surface of nanoparticles determines its enzyme-like activities, the use of other cerium-containing nanoparticles (for example, cerium fluoride, CeF_3_ NPs) should be considered. Previously, a comprehensive analysis of the biological activity of CeF_3_ NPs, in various models, confirmed the promise of this nanomaterial in various biomedical applications. For example, it has been demonstrated that CeF_3_ NPs protect living cells from exogenous H_2_O_2_ [14], and also protect freshwater planarian flatworms from the detrimental effects of X-ray irradiation as effective radioprotective agents [15]. These unique properties suggest that cerium-based or cerium-doped nanomaterials might provide the basis for new effective antimicrobial agents.

Another promising biologically active nanomaterial is photochromic tungsten oxide nanoparticles (WO_3_ NPs). WO_3_ NPs also demonstrate good antimicrobial properties, especially when illuminated, due to their high photocatalytic activity [16]. The antibacterial activity of tungsten oxide particles increases as their size decreases, and significantly depends on the degree of surface hydration. WO_3_ NPs have already been used successfully as a photocatalyst of the visible spectrum in the treatment and disinfection of wastewater. These particles have high biocompatibility and can be considered as antibacterial agents that are safe for human cells [17].

Combination therapies are an advanced approach whereby several therapeutic agents are used to effectively treat a condition, using synergistic or additive effects or reducing undesired effects. The principles of combination therapy are widely used in various fields of medicine, including HIV antiviral therapies [18], oncology [19] and the prevention of the development of drug resistance [20], e.g., bacterial resistance to antibiotics [21]. In this study, we combined the treatment of a model bacterium (*Escherichia coli*) with both CAP and various types of metal oxide/metal fluoride nanoparticles, to investigate the possible effect of two agents, for the development of advanced antimicrobial combination therapy.

## 2. Materials and Methods

### 2.1. Cold Atmospheric Argon Plasma Source

Low-temperature atmospheric pressure argon plasma was generated using a specially designed high-frequency current generator (frequency 50 kHz) with an amplitude of 4 kV (Appendix A). A plasma torch was created in an argon flow in a quartz tube 8 mm in diameter, using central and ring copper electrodes (Appendix A). The flow rate of argon was 2 L per min.

### 2.2. Model Organism

*E. coli* K-12 strain M61655 was used as a model organism. The bacterial species was cultured on Petri dishes (Merck, Darmstadt, Germany) with Luria–Bertani (LB) agar in super optimal medium with catabolic repressor (SOC) medium.

### 2.3. Nanoparticle Synthesis and Characterisation

CeO_2_ nanoparticles were synthesised using the precipitation method, according to a previously described protocol [22]. Briefly, 2.0 g of citric acid (Sigma-Aldrich, St. Louis, MO, USA, #251275) were mixed with 25 mL of 0.4 M aqueous cerium(III) chloride (Aldrich, #228931). The resulting solution was quickly poured, with stirring, into 100 mL of a 3 M ammonia solution (Khimmed, Moscow, Russia), kept for 2 h at ambient conditions and then boiled for 4 h. Then, the solution was cooled to room temperature and purified from precursors and by-products by precipitation and further redispersion.

CeF_3_ nanoparticles were synthesised by precipitation in alcohol media [23]. Briefly, 1.86 g of cerium(III) chloride heptahydrate (5 mM) (Aldrich, #228931) was dissolved in 15 mL of distilled water and added to 150 mL of isopropyl alcohol (Aldrich, #W292907). Hydrofluoric acid (20 mM) (Sigma-Aldrich, #30107) dissolved in 50 mL of isopropyl alcohol was added dropwise to the cerium salt solution, with vigorous stirring. The resulting white precipitate was filtered off and washed thoroughly several times with pure isopropyl alcohol. Then, the suspension was slightly dried, to form a pasty substance, and dispersed in 110 mL of distilled water using an ultrasonic bath. The resulting transparent colloidal solution was boiled for 5 min to remove alcohol residues. UV–visible absorption spectra of colloidal solutions were recorded in standard quartz cuvettes for liquid samples, using a UV5 Nano spectrophotometer (Mettler Toledo, Columbus, OH, USA). Transmission electron microscopy was performed using a Leo 912 AB Omega electron microscope (Carl Zeiss, Oberkochen, Germany) operating at an accelerating voltage of 100 kV. The particle size and zeta potential of nanoparticles were measured on a Benano 90 Zeta particle size analyser (Bettersize, Dandong, China).

Ultra-small hydrated WO_3_ nanoparticles were synthesised from tungstic acid in the presence of polyvinylpyrrolidone (PVP K-30, average mol. wt. 40,000) as a template, stabiliser and growth regulator, according to a method described earlier [24]. The electron microphotographs, UV/visible spectra and hydrodynamic radii distribution of the nanomaterials used in this study are presented in Appendix A. Zeta-potential values of the nanoparticles in distilled water were as follows: CeO_2_—−28.73 mV, CeF_3_—+39.22 mV, WO_3_—−8.11 mV (Appendix A). The following concentrations of the nanoparticles were used: CeO_2_—10^−4^ M, CeF_3_—10^−4^ M, WO_3_—2 × 10^−3^ M.

### 2.4. Inhibition Zone Test

The *E. coli* cell suspension (OD_625_ of 0.1) mixed with NP solution and the control sample sprayed onto LB agar plates were exposed to the plasma treatment. The plasma-emitting jet outlet was set at 1.0 cm above the agar layer containing the bacterial suspension. CAP was performed for two treatment periods (3 and 6 min). Cells were allowed to grow for 16–20 h at 37 °C and then inhibition zones were measured. The degree of inhibition was measured using ImageJ software (Version 1.53t).

### 2.5. Mutation Assay

The pal2-T plasmid was introduced into competent *E. coli* XL1-Blue cells (Evrogen, Moscow, Russia), according to the manufacturer’s instructions for chemical transformation: 0.5 μL of plasmid DNA was added to competent cells in 50 μL aliquots, left for 30 min on ice, subjected to heat shock (42 °C) for 60 s and transferred to ice for 10 min. Then, 300 μL of SOC was added to cells and incubated for 40 min at 37 °C, with shaking. The suspension with transformed cells was spread onto LB agar containing 1 mM isopropyl-β-D-thiogalactopyranoside (IPTG), 0.2 μg/mL 5-bromo-4-chloro-3-indolyl β-D-galactopyranoside (X-gal) (SibEnzyme LLC, Novosibirsk, Russia) and 100 μg/mL of ampicillin (SibEnzyme LLC, Novosibirsk, Russia) and incubated at 37 °C for 16–18 h. Blue-coloured cells were selected and placed in a tube with PBS to an OD_625_ of ~0.3. PBS was used instead of culture media during plasma treatment to eliminate the protective effects of media components. Then, 500 μL of the suspension was mixed with NP solutions and placed in 4-well plates for plasma treatment. The treatment procedure was performed as described previously. The exposure time was 3 min. The cultures were allowed to recover for 30 min at 37 °C, after which dilutions of 1:100 and 1:1000 were prepared and plated onto LB plates containing ampicillin, X-gal and IPTG, and incubated for 16–18 h at 37 °C and 1 h at 4 °C. Surviving colonies were then counted and photographs were taken. The results are represented as colony-forming units (CFUs).

### 2.6. RAPD PCR

The DNA isolation procedure was performed using an ExtractDNA Blood & Cells Kit (Evrogen, Moscow, Russia). DNA concentration was measured using a Qubit 4 Fluorometer (Thermo Fisher Scientific, Waltham, MA, USA); DNA samples were stored at −20 °C until use. A total of 20 random DNA oligonucleotide primers, synthesised by Evrogen (Russia), were used in the PCR reaction. Only six primers generated reproducible polymorphic DNA products. PCR amplification was performed in a 10 µL volume containing 20 ng of genomic DNA, 0.5 μL of primer (0.2 μM), 0.2 μL of dNTPs (2.5 mM), 1.0 μL of 10× buffer, 0.2 μL of Tersus polymerase (Tersus Plus PCR kit, Evrogen, Moscow, Russia) and sterile ddH_2_O. Amplifications were implemented in a DNA thermocycler (Bio-Rad, Hercules, CA, USA) and programmed to one cycle at 95 °C for 5 min, followed by 40 cycles at 94 °C for 1 min, 37 °C for 1 min and 72 °C for 2 min, with a final cycle of 72 °C for 5 min. PCR products were analysed by electrophoresis at 100 V for 45 min on 1.5% agarose gels in 1× TBE buffer. After electrophoresis, the RAPD patterns were visualised using a UV transilluminator (Vilber, Eberhardzell, Germany). RAPD markers were scored from the gels as DNA fragments that were present or absent in all lanes. The electropherograms were photographed.

Genomic template stability (GTS) was calculated for each primer (Table 1), using the formula: GTS (%) = (1 − a/n) × 100, where “a” is the number of polymorphic bands detected in each treated sample and “n” is the total number of bands in the control. Polymorphism observed in the RAPD profile included the disappearance of a normal band and the appearance of a new band in comparison with the control profile. To compare the sensitivity of the GTS parameter, changes in the values of this parameter were calculated as a percentage of the control (which was set to 100%).

### 2.7. Atomic Force Microscopy (AFM) Study of Bacteria after CAP Irradiation

Samples for the AFM characterisation were prepared as follows. A volume of 500 μL of *E. coli* suspension in PBS (OD_625_ of 0.1) with added NPs was placed in a 4-well culture plate, which was then exposed to plasma treatment. The plasma-emitting jet outlet was located 1.0 cm above the wells. The treatment duration was 3 and 6 min. After the treatment, the bacterial solution was centrifuged at 3000 rpm for 5 min, the supernatant was discarded, and bacterial sediment was resuspended in 50 μL of H_2_O. A 3 μL aliquot of the suspension was transferred onto a 1 × 1 cm piece of freshly-cleaved mica (TipsNano, Tallinn, Estonia) pretreated with a 1% polyethylenimine solution. AFM characterisation was performed using a SmartSPM-1000 atomic force microscope (AIST-NT, Co., Moscow, Russia). The tapping mode with a resonance frequency of 150–300 kHz was used in all scanning experiments.

### 2.8. Statistical Analysis

Each experiment included three parallel runs. The data obtained were processed using the SPSS software package (version 21.0 for Windows), featuring one-way analysis of variance (ANOVA). The results are expressed as the mean and standard deviation. 

## 3. Results

### 3.1. The Inactivation Effect of Plasma and NPs on E. coli

Photographs showing the enhanced antimicrobial effect of NPs in combination with plasma treatment are presented in Figure 1. ImageJ software (Version 1.53t) was used to quantify inactivation efficiency. The results are shown in Table 2 and demonstrate differences from irradiated control without NPs. Here, two treatment modes (3 and 6 min) and two concentrations of NPs (for CeO_2_ and CeF_3_—10^−5^ M and 10^−4^ M; for WO_3_—0.1 and 0.5 mg/mL) were used. A more pronounced effect was observed when using higher concentrations of all NPs investigated. The greatest effect was achieved in the case of 6 min of cold plasma treatment in combination with WO_3_ NPs. The use of WO_3_ NPs resulted in an additional 37% suppression of bacterial growth, in comparison with CAP treatment only. In the case of 3 min of CAP treatment, different types of NPs had the same effect. The plate surface without bacterial growth increased by an average of 24.89% when NPs were added to the bacterial suspension before treatment.

### 3.2. The Effect of NPs on Mutation in E. coli

To investigate the effects of NP pretreatment before plasma treatment on DNA in bacterial cells, a mutation assay was used. The impact on the DNA structure was tested using the pAL2-T plasmid. This plasmid carries the genetic information for two enzymes, β-lactamase, which confers ampicillin resistance, and b-galactosidase (lacZ), which can convert colourless X-Gal to a blue dye. When competent *E. coli* cells are transformed with the plasmid, the growth of colonies on ampicillin-containing LB agar indicates successful transformation. Blue colonies indicate the presence of functional b-galactosidase, whereas in white colonies, a functional b-galactosidase is absent (Figure 2). An experimental group without treatment (negative control group) did not contain white colonies and is not represented on the graph, for clarity. Figure 2 shows an overall decrease in the number of colonies after CAP treatment and an even greater reduction after pretreatment with NPs. It should be noted that NP pretreatment itself, without further CAP treatment, did not influence the number of colonies (Figure 2a). The CFU-counting method was used to quantify bacterial inactivation efficiency by plasma treatment. The CFU number of bacteria after the treatment decreased by, on average, 69.26%; in the case of combined treatment (NPs + CAP), the reduction was by 94.14%. The number of blue and white colonies determined after incubation is shown in Figure 2. The most pronounced effect on mutation appearance was caused by the combination of CAP and CeF_3_ NPs, resulting in approximately 20% colourless colonies. For CeO_2_, colourless colonies accounted for 10.53%; for WO_3_, the figure was 9.09%.

### 3.3. RAPD PCR

The RAPD-PCR profile of *E. coli* showed a total of 43 reproducible bands amplified with six different primers (Table 3). Genomic template stability (GTS) is a percentage value that reflects the PCR amplification profile changes of a test sample relative to a control sample. The total GTS value was calculated in each of the RAPD profiles generated using six selected screening primers. GTS% was calculated as follows [25]: GTS% = (1 − a/n) × 100, where (a) is the DNA polymorphism profile in each polluted site and (n) is the total number of bands in the control. Polymorphism included the disappearance of a normal band and the appearance of a new band in comparison with the control profiles. The percentage of polymorphism ranged from 100% to 41.86%, as shown in Figure 3. 

Groups treated with NPs only showed no differences from the control. The GTS of CAP-treated groups without NPs was 69.77% after 3 min of treatment and 88.37% after 6 min of treatment. Treatment with both NPs and plasma, in the case of CeO_2_ and CeF_3_ NPs, had virtually no effect on GTS, but a significant difference was observed after 3 min of CAP treatment with WO_3_-NPs. Here, the GTS value was 41.86%.

### 3.4. Atomic Force Microscopy

Plasma affects the bacterial cell walls in a destructive way, leading to their deformation (Figure 4). This effect is enhanced with increasing exposure time. The destruction of the cell walls and the aggregation of bacterial cells can be clearly observed. The addition of cerium- or tungsten-containing oxide nanoparticles enhanced the effect of the plasma. The most obvious changes and the strongest effect were observed when using CeF_3_ NPs with CAP: the cell wall was completely destroyed, and the cell content was released. CeO_2_ NPs acted in the same way, but the effect was less pronounced. WO_3_ NPs did not influence the bacterial cell structure in such a destructive way.

## 4. Discussion and Conclusions

The study showed that a combined treatment of bacterial colonies with cold argon plasma and nanoparticles had a pronounced antimicrobial effect, exceeding that of any of the components alone. Metal oxide nanoparticles are known to possess antimicrobial activity [26], and WO_3_ was previously shown to inhibit the growth of gram-positive and gram-negative bacteria and viruses, and to have a damaging effect on cell membranes [27,28]. CeO_2_ has also been reported previously to be an effective agent against a broad spectrum of pathogens [29]. Information on the antibacterial properties of CeF_3_ NPs, however, continues to be absent. The enhancement of NPs’ antimicrobial properties by CAP treatment paves the way for the effective use of this combination to heal chronic wounds, where CAP is already being used efficiently [30].

In this study, different combinations of the argon plasma treatment with inorganic redox-active NPs were tested. Bacterial growth inhibition tests showed a synergistic effect for the combination of CAP with all types of nanoparticles tested, exceeding the effect of CAP alone. The blue–white colony test showed that the CAP-WO_3_ combination was the most effective against colony growth. The CAP-CeF_3_ combination, however, had the most pronounced mutagenic effect on plasmid DNA, producing the largest percentage of white colonies. CAP-CeF_3_ and, to a lesser degree, CAP-CeO_2_ were responsible for the most pronounced cell wall damage in AFM studies, while CAP-WO_3_ had a more moderate impact. For the first time, the antimicrobial and DNA-damaging effect of CeF_3_ nanoparticles has been shown; previously, these NPs were reported as being antioxidant agents only [23]. Finally, the CAP-WO_3_ combination demonstrated a strong mutagenic effect on bacterial genomic DNA, while the impact of the other combinations did not significantly differ from that of CAP alone.

The key factor determining the greater impact of CAP-CeF_3_ on cell walls and plasmid DNA was possibly the positive charge of NPs. Surface charge has been shown previously to be crucial for the cytotoxic effect of CeO_2_ nanoparticles [31]. The zeta potential measurements presented above show that CeF_3_ NPs possess a substantial positive surface charge, whereas CeO_2_ and WO_3_ are charged negatively. Thus, CeF_3_ nanoparticles can be electrostatically adsorbed on a negatively charged bacterial cell wall, providing a higher cytotoxicity for the CAP-CeF_3_ combination. It is assumed that the mutagenic effect is responsible for the penetration of NPs into bacterial cells. While genomic DNA is supercoiled and protected from interactions with NPs by HU and other nucleoid-associated proteins [32], plasmid DNA can be more accessible for electrostatic interactions with positively charged CeF_3_. At the same time, the mutagenic effect of negatively charged CeO_2_ and WO_3_ NPs in combination with CAP on genomic DNA is governed by other factors, which may be more prominent for WO_3_ because of its higher concentration in the experiment.

In general, the combination of cold argon plasma treatment and metal oxide/metal fluoride nanoparticles has demonstrated a pronounced synergistic antimicrobial effect, and it can be recommended as the subject of further studies, to develop novel wound-healing therapies and disinfection methods.

## Figures and Tables

**Figure 1 biomedicines-11-02780-f001:**
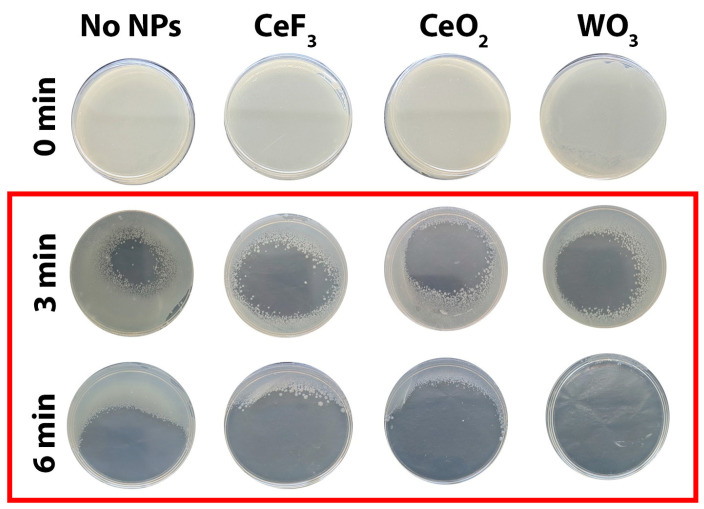
*E. coli* growth inhibition zones after CAP treatment. Control plate (0 min)—no CAP treatment, no NPs.

**Figure 2 biomedicines-11-02780-f002:**
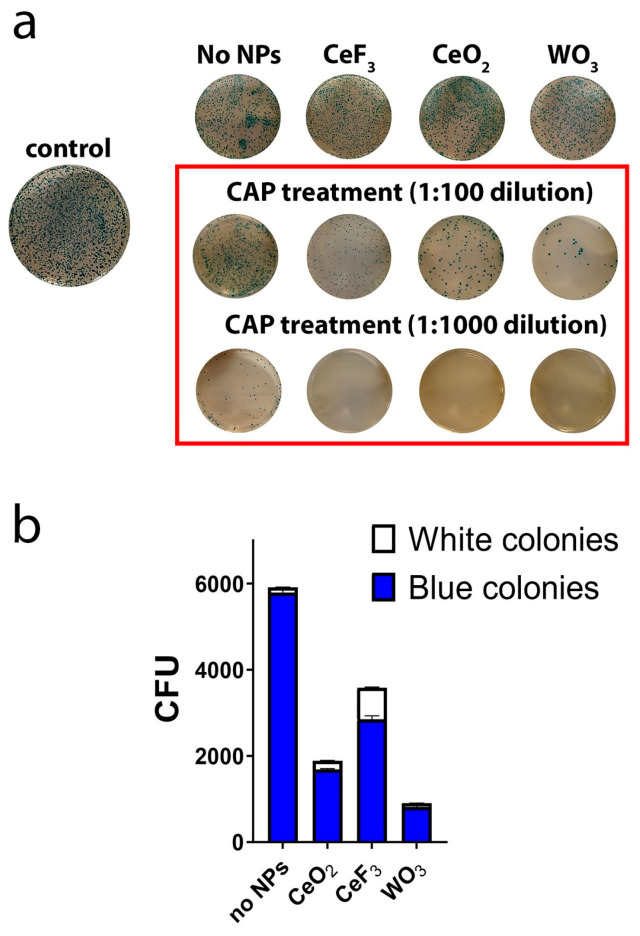
Antimicrobial and mutagenic effects of CAP in combination with CeO_2_, CeF_3_ and WO_3_ NPs, as assessed using blue/white colouring assay on *E. coli* (**a**). Blue/white colony ratio after CAP treatment (**b**).

**Figure 3 biomedicines-11-02780-f003:**
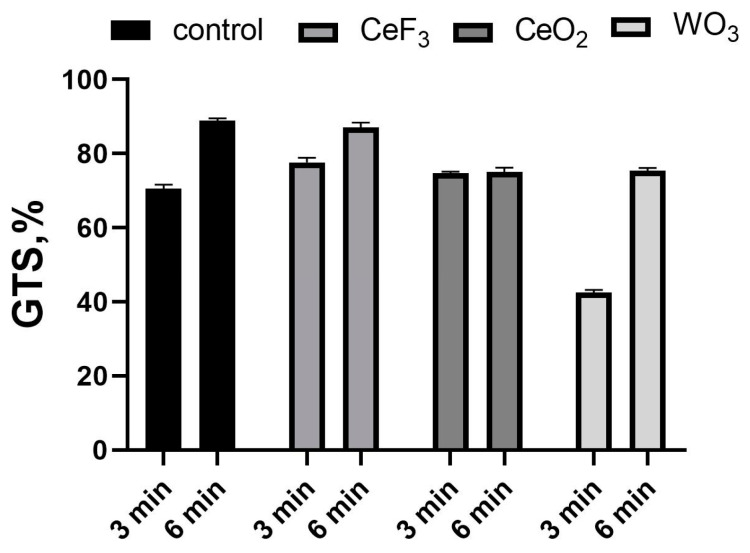
Genomic DNA template stability (GTS) of *E. coli*.

**Figure 4 biomedicines-11-02780-f004:**
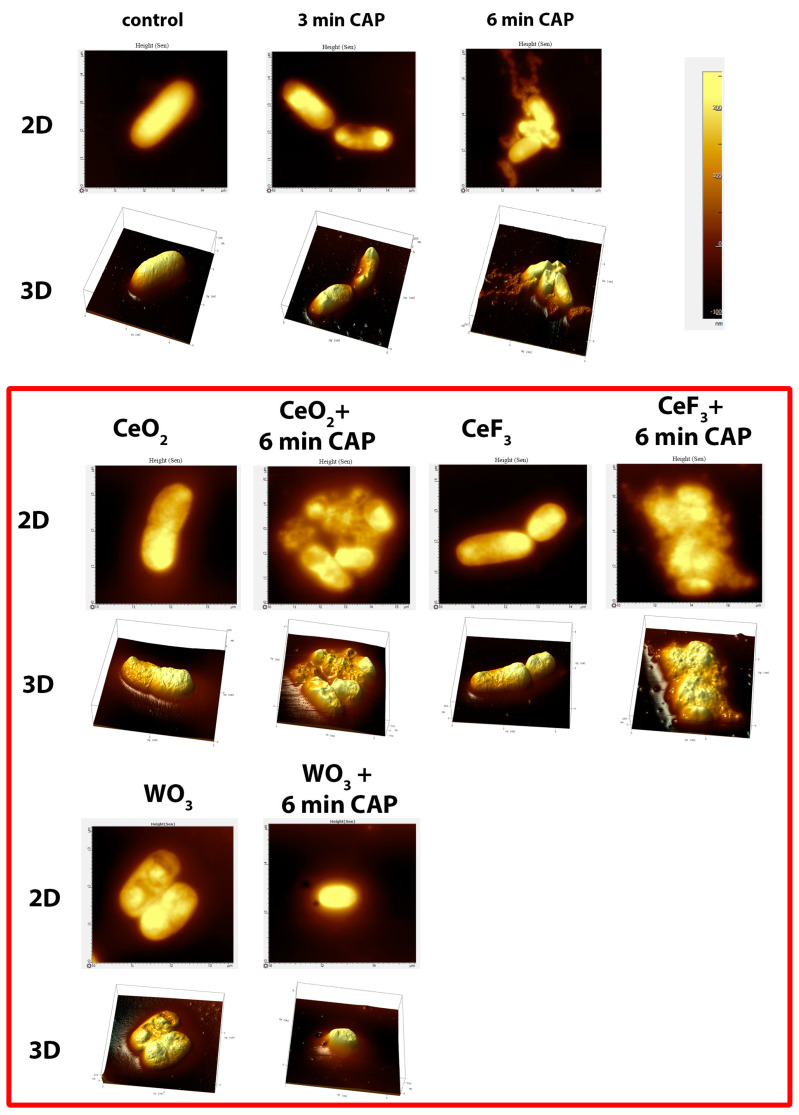
Representative AFM images of bacteria after treatment with CAP, nanoparticles or combinations of these. The scan area was 30–100 µm^2^; a typical z-axis scale is 200 nm.

**Table 1 biomedicines-11-02780-t001:** Set of primers used for RAPD-PCR analysis.

Primer	Nucleotide Sequence (5′→3′)
OPB5	TGCGCCCTTC
OPB7	GGTGACGCAG
OPV14	GGTCGATCTG
OPX7	GAGCGAGGCT
OP-B04	GATGACCGCC
OP-Q18	GGGAGCGAGT

**Table 2 biomedicines-11-02780-t002:** The effect of CeF_3_, CeO_2_ and WO_3_ nanoparticles on the size of the no-growth area of microorganisms.

NP Type	Bacterial Growth Suppression with the Combination of CAP and NPs (in Comparison with the Control Treated with CAP Only)	
	3 min treatment	6 min treatment
CeF_3_	28.12% ± 1.41	17.05% ± 0.85
CeO_2_	24.35% ± 1.12	24.73% ± 1.19
WO_3_	22.21% ± 0.91	37.04% ± 1.84

**Table 3 biomedicines-11-02780-t003:** Changes in the total number of bands in the control and treated samples of *E. coli*. The first value reflects the appearance of new bands; the second the disappearance of control bands. n—total number of bands in the control.

Primer	Group											
	Control	No NPs		CeF_3_			CeO_2_			WO_3_		
		3	6	n	3	6	n	3	6	n	3	6
OPB5	8	0; 4	0; 6	0; 0	0; 3	0; 0	0; 0	0; 4	0; 1	0; 0	0; 6	0; 3
OPB7	8	0; 0	0; 0	0; 0	0; 3	0; 2	0; 0	0; 3	0; 2	0; 0	0; 4	0; 3
OPV14	7	0; 3	0; 1	0; 0	0; 0	0; 1	0; 0	0; 0	0; 4	0; 0	0; 5	0; 3
OPX7	6	0; 5	2; 1	0; 0	1; 0	0; 0	0; 0	1; 0	0; 1	0; 0	0; 2	0; 0
OPB4	7	0; 1	0; 0	0; 0	0; 1	0; 1	0; 0	0; 1	0; 1	0; 0	0; 3	0; 0
OPQ18	7	0; 0	0; 0	0; 0	0; 2	0; 2	0; 0	0; 2	0; 2	0; 0	0; 5	0; 2
Total bands	43											
Polymorphic bands		13	5	0	10	6	0	11	11	0	25	11

## Data Availability

Data is contained within the article or Appendix A.

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
