# Peer review of "Synergistic Antimicrobial Effect of Cold Atmospheric Plasma and Redox-Active Nanoparticles"

_biomedicines, 2023, doi:10.3390/biomedicines11102780_

Round 1

Reviewer 1 Report

Manuscript entitled “Synergistic antimicrobial effect of cold atmospheric plasma and redox-active nanoparticles” by Ermakov and colleagues in Biomedicine is good piece of work. It is hot topic and timely topic. Research manuscript is well written and scientifically sound. I recommend it as it is without any further revision.   

Author Response

We are very grateful for the reviewers’ comments aimed at improving our paper. We have thoroughly revised the manuscript in accordance with the reviewers’ comments. We have carefully checked all the points and we have tried to address all the questions and suggestions.

Reviewer #1

General comment:

Manuscript entitled “Synergistic antimicrobial effect of cold atmospheric plasma and redox-active nanoparticles” by Ermakov and colleagues in Biomedicine is good piece of work. It is hot topic and timely topic. Research manuscript is well written and scientifically sound. I recommend it as it is without any further revision.   

Discussion: We thank the reviewer for the positive evaluation of our work.

Reviewer 2 Report

The following comments are given to the authors for revision 

1.       Introduction

Cold argon plasma (CAP) is ionised gas cooled to a temperature comparable to a physiological value (30-40oC), whereas plasma generated under standard environmental conditions has a temperature of 3,000-5,000oC, having a devastating effect on biological objects [1]

Is it correct statement?  Is the plasma generated having the temperature unit 3000-5000oC?. The statement need to be corrected for clear understanding

Its microbiocidal effect is achieved by the generation of reactive species such as ROS and RNS, charged particles, UV photons and other factors. Ref 3 and 4

The authors need to resentence the above statement (In initial reading it gives the meaning of microbial species?) such as ROS and RNS – First time expand the abbreviations, wherever applicable) LB agar, CeO2; SOC etc..1

Para 3: It has been demonstrated that cerium oxide NPs can destroy bacterial biofilms  

Need to mention as cerium oxide (CeO2) in the first attempt instead giving in the 3rd para of 11th reference.

Chemical/ Instrument identification: UV5 Nano spectrophotometer (METTLER TOLEDO).

For chemicals, some places mentioned as sigma Aldrich in some Aldrich with code numbers. Recommended to follow uniformity as: Company name, City and Country.  Follow the same procedure uniformly for all the chemicals and Instruments, wherever applicable.

2.4. Inhibition zone test:

Cells were allowed to grow for 16-20 h at 370C before inhibition zones were measured

Before inhibition zones were measured or after inhibition zones were measured?

Define why the antibiotic ampicillin containing LB agar plates used and what was the ampicillin concentration? In 2.5 mutation assay.

3.2. The effect of NPs on mutation in E. coli

Is it  b-lactamase or β –lactamase?

Need minor revision 

Author Response

Reviewer #2

Issue 1.       Cold argon plasma (CAP) is ionised gas cooled to a temperature comparable to a physiological value (30-40oC), whereas plasma generated under standard environmental conditions has a temperature of 3,000-5,000 oC, having a devastating effect on biological objects [1]. Is it correct statement?  Is the plasma generated having the temperature unit 3000-5000oC?. The statement need to be corrected for clear understanding

Discussion: We thank the reviewer for this comment. In our device, the plasma does not have a temperature of 3,000-5,000 oC. The plasma temperature at the nozzle exit does not exceed 40 °C. We have clarified and modified the description of the term «cold plasma».

Changes made in the manuscript:   Cold argon plasma (CAP) is ionised gas which temperature is comparable to a physiological value (30-40oC), whereas so called thermal plasma used in industrial applications has a temperature of 3,000-5,000oC, having a devastating effect on biological objects [1].

Issue 2.      Its microbiocidal effect is achieved by the generation of reactive species such as ROS and RNS, charged particles, UV photons and other factors. Ref 3 and 4. The authors need to resentence the above statement (In initial reading it gives the meaning of microbial species?) such as ROS and RNS – First time expand the abbreviations, wherever applicable) LB agar, CeO2; SOC etc..1

Discussion: We thank the reviewer for this comment. We corrected this sentence and introduced explanations of all the abbreviations used (LB agar, CeO2; ROS, RNS, SOC etc).

Changes made in the manuscript:   We changed this sentence as follows: «Its microbiocidal effect is achieved via damaging and killing bacteria by reactive oxygen species (ROS), reactive nitrogen species (RNS), charged particles, UV photons, generated in the plasma flow».

Abbreviations explained in the manuscript: Luria-Bertani (LB) agar, Super optimal medium with catabolic repressor (SOC medium), reactive oxygen species (ROS), reactive nitrogen species (RNS), cerium oxide nanoparticles (CeO2).

Issue 3.      Para 3: It has been demonstrated that cerium oxide NPs can destroy bacterial biofilms. Need to mention as cerium oxide (CeO2) in the first attempt instead giving in the 3rd para of 11th reference.

Discussion: We thank the reviewer for this comment.

Changes made in the manuscript:   We corrected the sentence according to the Reviewer’s comment.

Issue 4.      Chemical/Instrument identification: UV5 Nano spectrophotometer (METTLER TOLEDO). For chemicals, some places mentioned as sigma Aldrich in some Aldrich with code numbers. Recommended to follow uniformity as: Company name, City and Country.  Follow the same procedure uniformly for all the chemicals and Instruments, wherever applicable.

Discussion: We thank the reviewer for this comment.

Changes made in the manuscript:   We checked and corrected the information on the suppliers of the chemicals and instruments.

Issue 5.      2.4. Inhibition zone test: Cells were allowed to grow for 16-20 h at 370C before inhibition zones were measured. Before inhibition zones were measured or after inhibition zones were measured?

Discussion: We thank the reviewer for this comment. We corrected this sentence as follows.

Changes made in the manuscript:   Cells were allowed to grow for 16-20 h at 370C and then the inhibition zones were measured.

Issue 6.      Define why the antibiotic ampicillin containing LB agar plates used and what was the ampicillin concentration? In 2.5 mutation assay.

Discussion: We thank the reviewer for this comment. We used a special plasmid carrying the ampicillin resistance gene and the β-galactosidase gene to select the desired clones. Ampicillin resistance allowed to select bacterial colonies with this plasmid present in the cells. The final concentration of ampicillin in LB agar was 100 µg/mL.

Changes made in the manuscript:  We added the following text to the section 2.5: «The suspension with transformed cells was spread onto LB agar containing 1 mM isopropyl-β-D-thiogalactopyranoside (IPTG), 0.2 μg/ml 5-bromo-4-chloro-3-indolyl β-D-galactopyranoside (X-gal)(SibEnzyme LLC, Novosibirsk, Russia) and 100 μg/ml of ampicillin (SibEnzyme LLC, Novosibirsk, Russia) and incubated at 370C for 16-18 h».

Issue 7. 3.2. The effect of NPs on mutation in E. coli. Is it b-lactamase or ?

Discussion: We thank the reviewer for this comment. Exactly, it is β –lactamase.

Changes made in the manuscript:   We corrected the typo.

Reviewer 3 Report

In this manuscript, the authors provided a strong antimicrobial and DNA-damaging effect of CeF3 nanoparticles has been demonstrated. The studies were thought out and executed well. However, we have a few questions that should be addressed prior to publication.

1)      The electric potential of the nanoparticles was mentioned in the main text, please supplement the potential figure and data of nanoparticles.

2)      Whether nanoparticles are safe in vivo? Please supplement the data to provide the evidence.

3)      How did the authors decide on the ratio of CAP into nanoparticles?

Minor editing of English language required.

Author Response

We are very grateful for the reviewers’ comments aimed at improving our paper. We have thoroughly revised the manuscript in accordance with the reviewers’ comments. We have carefully checked all the points and we have tried to address all the questions and suggestions.

Reviewer #3

General comment:

In this manuscript, the authors provided a strong antimicrobial and DNA-damaging effect of CeF3 nanoparticles has been demonstrated. The studies were thought out and executed well. However, we have a few questions that should be addressed prior to publication.

Discussion: We thank the reviewer for the positive evaluation of our work.

Issue 1. The electric potential of the nanoparticles was mentioned in the main text, please supplement the potential figure and data of nanoparticles.

Discussion: We thank the reviewer for this comment.

Changes made in the manuscript:  We provided the zeta potential values in the Supplementary file (Figure S4).

Issue 2. Whether nanoparticles are safe in vivo? Please supplement the data to provide the evidence.

Discussion: We thank the reviewer for this comment. In vivo experiments are far beyond the scope of our paper which deals with in vitro experiments only. Currently, the following limited information is available on safety of CeO2, CeF3 and WO3 nanoparticles use:

Cerium species have a long history of biomedical tests [Shcherbakov, A.B.; et al. Elsevier: Amsterdam, The Netherlands, 2020; pp. 279–358.]. In toxicity studies of CeO2 nanoparticles, no adverse effects were reported at concentrations well above therapeutic levels as reported after oral administration of CeO2 nanoparticles at a dose of 1000 mg/kg body weight for 14 days.  (Lee, J.; et al Nanotoxicology 2020, 14, 696–710). There is a large amount of data on the biosafety of these cerium oxide nanoparticles during long-term exposure (Yokel, R.A.et al Nanomedicine 2013, 9, 398–407; Yokel, R.A.; et al Toxicol. Sci. 2012, 127, 256–268; Yokel, R.A.; et al. Environ. Sci. Nano 2014, 1, 406.). The world's first acute oral poisoning with cerium dioxide nanoparticles with precise exposure concentrations was recently reported (Lu Y.Q. Environ Toxicol Pharmacol. 2021;82:103560). The patient recovered well after treatment with cerium removal and restoration of clotting factor activity without any side effects.

Cerium species have relatively low toxicity per os: LD50 4200 mg kg-1 for cerium nitrate (female rats) and LD50 1178 mg kg-1 (female mice) [Bruce, D.W.; et al. Toxicol. Appl. Pharmacol. 1963, 5, 750–759.]. These values are close to the toxicity parameters of table salt (LD50 3000 mg kg-1 for rats [Feldman, S.R. JohnWiley & Sons, Inc.: Hoboken, NJ, USA, 2011.]). The low risk associated with the oral administration of cerium species is supported by their low absorption, with more than 95% of the species being excreted in the faeces [Morganti, J.B.; Gen. Pharmacol. Vasc. Syst. 1978, 9, 257–261.]. Moreover, orally administered lanthanide compounds at low doses have even been reported to have beneficial effects for laboratory and farm animals [Aquilina, G.; EFSA J. 2013, 11, 3206.] resulting in a significant increase in their immune level. According to the most recent report, orally administered cerium species have great potential for use in dental practise as they prevent inflammatory changes in periodontal tissues in obese humans with generalized catarrhal gingivitis [Skrypnyk, M.; Lett. Appl. NanoBiosci. 2019, 8, 754–761.]. Further implementation of the current research requires the translation of in vitro experiments to preclinical and clinical trials, as well as conversion of the cellular half maximal inhibitory concentration IC50 (mmol/L) into corresponding toxicity values for animals and humans (mg/kg), such as median lethal dose LD50, no observed adverse effect level NOAEL and no observed effect level NOEL. According to a prediction model recommended by the International Workshop on In Vitro Methods for Assessing Acute Systemic Toxicity [NIH Publication No. 01-4499; NC, USA, 2001.], LD50 values (in mmol/kg) can be estimated from IC50x values (in mmol/L) using the following relation: log (LD50) = 0.435 _ log (IC50x) + 0.625. For cerium (III) chloride, IC50 = 5–10 mM (see above), thus LD50 is estimated to be 8.5 –11.5 mmol/kg or 2100–2800 mg/kg, which is in good agreement with experimentally measured values of 2800 mg/kg (male and female rats per os). NOEL and NOAEL are also assessed prior to the initiation of human trials, in order to establish a safe clinical starting dose. For cerium nitrate, NOEL was reported to be 110 mg/kg/day and NOAEL was reported to be 330 mg/kg/day (47-day, rats per os) [Shin, S.-H.; Saf. Health Work 2019, 10, 409–419.]. For a more accurate transfer of drug doses from animal to human trials, the body surface area (BSA) normalization method has been suggested [Reagan-Shaw, S.; FASEB J. 2008, 22, 659–661]. BSA normalization was advocated for translation from animals to humans in phase I and phase II clinical trials: HED = Animal dose_Animal Km Human Km (here, HED—human equivalent dose (mg/kg), Km—surface area factor). For rats, Km= 6 and, for humans, Km= 37 (adults) or 25 (children). Taking into account the above considerations, the NOEL value for water-soluble cerium species for adult humans can be estimated as 18 mg/kg/day or about 1 g/day (for the body weight ~60 kg). Common oral rinses contain ~0.05% (230 ppm F) NaF, and thus the accidental daily swallowing of 10 mL rinse solution is equal to the ingestion of ~2.3 mg of fluoride [Marinho, V.C.; Cochrane Database Syst. Rev. 2016, 7, CD002284], while the probable toxic dose (PTD) of fluoride was estimated to be 5 mg/kg.

There are quite a few works on the analysis of the toxicity of WO3 NPs. We have previously shown the absence of toxicity on normal and cancer cells at concentrations below 200 µg/mL (B. Han, A. L. Popov, T. O. Shekunova, D. A. Kozlov, O. S. Ivanova, A. A. Rumyantsev, A. B. Shcherbakov, N. R. Popova, A. E. Baranchikov, V. K. Ivanov, "Highly Crystalline WO3 Nanoparticles Are Nontoxic to Stem Cells and Cancer Cells", Journal of Nanomaterials, vol. 2019, Article ID 5384132, 13 pages, 2019.)

Changes made in the manuscript:   No changes were made.

Issue 3. How did the authors decide on the ratio of CAP into nanoparticles?

Discussion: CAP irradiation duration was selected experimentally taking into account the size of the inhibition zone. Nanoparticles concentrations were chosen based on the previously reported toxicity data [B. Han, et al Journal of Nanomaterials, vol. 2019, Article ID 5384132, 13 pages, 2019.; Popov A. et al. Nanosystems: Physics, Chemistry, Mathematics, 2021, 12 (3), 1–7, Popov A. et al. Materials Science & Engineering C, 2020,108, 110494; Ermakov A. et al,; Materials Science and Engineering: C 2019, 104, 109924; Popov A. et al. Nanomaterials 2022, 12(17), 3034].

Changes made in the manuscript:   No changes were made.

Reviewer 4 Report

Dear Authors,

I find this a proper and complete work according to your previous studies and recommend for it to be published.

Author Response

We are very grateful for the reviewers’ comments aimed at improving our paper. We have thoroughly revised the manuscript in accordance with the reviewers’ comments. We have carefully checked all the points and we have tried to address all the questions and suggestions.

Reviewer #4

General comment:

General comment Dear Authors,

I find this a proper and complete work according to your previous studies and recommend for it to be published.

Discussion: We thank the reviewer for the positive evaluation of our work.
